# PROMPT-FREE DIFFUSION: TAKING "TEXT" OUT OF TEXT-TO-IMAGE DIFFUSION MODELS

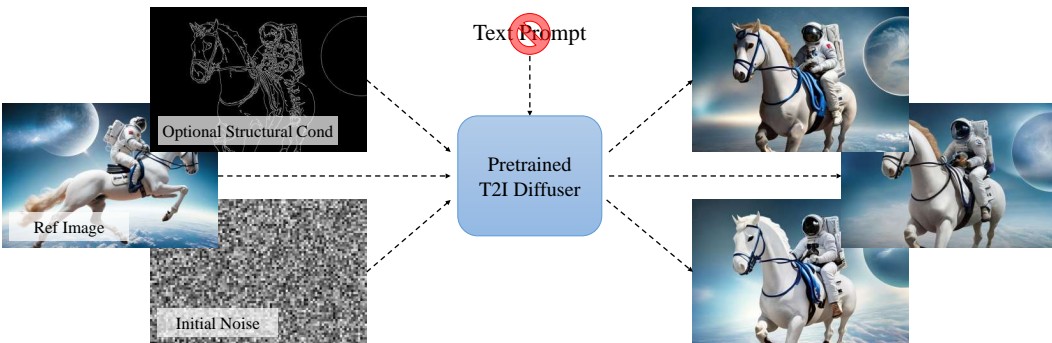

Figure 1: Given a pre-trained Text-to-Image (T2I) diffusion model, **Prompt-Free Diffusion** modifies it to intake a reference image as "context", an *optional* image structural conditioning (i.e. canny edge in this case), and an initial noise. No text prompt is needed: instead, the reference image governs the semantics and appearance, and the optional conditioning provides additional control over the structure. Note that the generated image instances are **precisely controlled** (with only **small variations**, as larger variations would mean less control precision), to faithfully reflect the desired style (reference image) and structure (conditioning).

## ABSTRACT

Text-to-image (T2I) research has grown explosively in the past year, owing to the large-scale pre-trained diffusion models and many emerging personalization and editing approaches. Yet, **one pain point persists: the text prompt engineering**, and searching high-quality text prompts for customized results is more art than science. Moreover, as commonly argued: "an image is worth a thousand words" - the attempt to describe a desired image with texts often ends up being ambiguous and cannot comprehensively cover delicate visual details, hence necessitating more additional controls from the visual domain. In this paper, we take a bold step forward: taking "Text" out of a pre-trained T2I diffusion model, to reduce the burdensome prompt engineering efforts for users. Our proposed framework, **Prompt-Free Diffusion**, relies on **only visual inputs to generate new images**: it takes a reference image as "context", an optional image structural conditioning, and an initial noise, with absolutely no text prompt. The core architecture behind the scene is **Se**mantic Context **Encoder** (**SeeCoder**), substituting the commonly used CLIP-based or LLM-based text encoder. The reusability of SeeCoder also makes it a convenient drop-in component: one can also pre-train a SeeCoder in one T2I model and reuse it for another. Through extensive experiments, Prompt-Free Diffusion is experimentally found to (i) outperform prior exemplar-based image synthesis approaches; (ii) perform on par with state-of-the-art T2I models using prompts following the best practice; and (iii) be naturally extensible to other downstream applications such as anime figure generation and virtual try-on, with promising quality. Our code and models will be open-sourced.

## 1 INTRODUCTION

The high demand for personalized synthetic results has promoted several text-to-image (T2I) related technologies, including model finetuning, prompt engineering, and controllable editing, to a new

level of importance. A considerable number of recent works such as Ruiz et al. (2023); Lu et al. (2023); Xu et al. (2022); Hu et al. (2021); Zhang & Agrawala (2023); Mou et al. (2023); Huang et al. (2023); Guo et al. (2023); Goel et al. (2023); Wang et al. (2023) also emerge to address personalization tasks from different angles. By far, it remains an open question of what is the most convenient way to achieve such personalization. One straightforward approach is to fine-tune a T2I model with exemplar images. Personalized tuning techniques Ruiz et al. (2023); Lu et al. (2023); Hu et al. (2021); Zhang et al. (2023) such as DreamBooth Ruiz et al. (2023) have shown promising quality by finetuning model weights. Their downsides are yet obvious: finetuning a model remains to be resource-costly for average users, despite the growing effort of more efficient tuning Hu et al. (2021). Prompt-engineering Witteveen & Andrews (2022); Guo et al. (2023) serves as a lite alternative for personalizing T2I models. It has been widely adopted in the industry due to its excellent cost margin, *i.e.* improving output quality from almost zero cost. Nevertheless, searching high-quality text prompts for customized results is more art than science. Moreover, as commonly argued: "an image is worth a thousand words" - the attempt to describe a desired image with texts often ends up being ambiguous, and cannot comprehensively cover delicate visual details.

To better handle personalized needs, several T2I controlling techniques and adaptive models are proposed Zhang & Agrawala (2023); Mou et al. (2023); Huang et al. (2023); Wang et al. (2023); Goel et al. (2023). Representative works such as ControlNet Zhang & Agrawala (2023), T2I-Adapter Mou et al. (2023), *etc.* proposed the adaptive siamese networks that take user-customized structural conditionings (*i.e.* canny edge, depth, pose, *etc.*) as generative guidances in additional to prompts. Such an approach has become one of the most popular among downstream users since it: a) disentanglement of structure from content enables more precise controls over results than prompt, b) these plug-and-run modules are reusable to most T2I models that save users from extra training stages.

Do methods like ControlNet Zhang & Agrawala (2023) *etc.* resolve all the challenges in personalized T2I? Unfortunately, the answer is not quite. For example, it still faces the following issues: a) the current approach meets challenges in generating user-designated textures, objects, and semantics; b) searching prompt-in-need sometimes is problematic and inconvenient; and c) prompt engineering for quality purposes is still required. In conclusion, all these issues come from the fundamental knowledge gap between vision and language. Captions alone may not provide a comprehensive representation of all visual cues, and providing structural guidance does not eliminate this problem.

To overcome the aforementioned challenges, we introduced the novel *Prompt-Free Diffusion*, replacing the regular prompts input with reference images. Speaking with more details, we utilize the newly proposed *Semantic Context Encoder (SeeCoder)*, in which the pixel-based images with arbitrary resolutions can be auto-transformed into meaningful visual embeddings. Such embeddings can represent low-level information, such as textures, effects, *etc.*, and high-level information, such as objects, semantics, *etc.* We then use these visual embeddings as the conditional inputs of an arbitrary T2I model, generating high-performing customized outputs on par with the current state-of-the-art. One may notice that our Prompt-Free Diffusion shares similar goals as exemplar-base image generation Park et al. (2019); Zhang et al. (2020); Zhou et al. (2021a); Choi et al. (2021); Lee et al. (2022) and image-variation Ramesh et al. (2022); Xu et al. (2022). But it stands out from prior approaches with quality and convenience. Explicitly speaking, SeeCoder is *reusable* to most open-sourced T2I models in which one can easily convert the T2I pipeline to our Prompt-Free pipeline without much effort. While this is mostly infeasible in prior works (*i.e.* exemplar-base generation, image-variation) in which they require either specific models for specific domains or finetuning models deviated from T2I purpose.

Our main contributions are concluded in the following:

- We proposed *Prompt-Free Diffusion*, an effective solution generating high-quality images utilizing text-to-image diffusion models without text prompts.

- Empowered by the reusability of *Semantic Context Encoder (SeeCoder)*, the proposed Prompt-Free property can be available in many other existing text-to-image models without extra training, creating a convenient pipeline for personalized image generation.

- Our method can be extended to many downstream applications with competitive quality, such as exemplar-based virtual try-on, and anime figure generation.

## 2 RELATED WORKS

### 2.1 TEXT-TO-IMAGE DIFFUSION

Diffusion models (DM) Sohl-Dickstein et al. (2015); Ho et al. (2020); Song et al. (2020); Dhariwal & Nichol (2021); Ramesh et al. (2022); Saharia et al. (2022); Rombach et al. (2022) nowadays is the *de facto* workhorsse for Text-to-Image (T2I) generation. Diffusion-based T2I models Nichol et al. (2021); Ramesh et al. (2022); Saharia et al. (2022); Rombach et al. (2022); Liu et al. (2023); Gu et al. (2022) generate photorealistic images via iterative refinements. GLIDE Nichol et al. (2021) introduced a cascaded diffusion structure and utilized classifier-free guidance Ho & Salimans (2022) for image generation and editing. DALL-E2 Ramesh et al. (2022) proposed a model with several stages, encoding text with CLIP Radford et al. (2021), decoding images from text encoding, and upsampling them from $64^2$ to $1024^2$. Imagen Saharia et al. (2022) discovered that scaling up the size of the text encoder Devlin et al. (2018); Raffel et al. (2020); Radford et al. (2021) improves both sample fidelity and text-image alignment. VQ-Diffusion Gu et al. (2022) learned T2I diffusion models on the discrete latent space of VQ-VAE Van Den Oord et al. (2017). The popular latent diffusion model (LDM, *i.e.* Stable Diffusion) Rombach et al. (2022) investigated the diffusion process over the latent space of pre-trained encoders, improving both training and sampling efficiency without quality degradation. Versatile Diffusion Xu et al. (2022) further enables LDM across multimodal generation and natively supports image variation.

Although generative adversarial networks (GANs) Goodfellow et al. (2020); Brock et al. (2018); Karras et al. (2020)) and autoregressive (AR) models Brown et al. (2020a); Radford et al. (2018; 2019) show T2I capability to some extent, most of them Zhang et al. (2017; 2018); Xu et al. (2018); Zhu et al. (2019); Tao et al. (2020); Chen et al. (2020) generate images on specific domains instead of open-world text sets. With the advances of large-scale language encoder Radford et al. (2021); Raffel et al. (2020); Brown et al. (2020b), GANs Zhou et al. (2021b); Tao et al. (2023); Sauer et al. (2023); Kang et al. (2023) and AR models Ramesh et al. (2021); Ding et al. (2021); Gafni et al. (2022); Yu et al. (2022) recently start to handle generation with arbitrary texts with promising qualities too.

### 2.2 EXEMPLAR-BASED GENERATION

Given an exemplar image, exemplar-based generation Park et al. (2019); Zhang et al. (2020); Zhou et al. (2021a); Zhan et al. (2021); Guo et al. (2022); Zhan et al. (2022); Liu et al. (2022); Seo et al. (2022); Bhunia et al. (2023); Zhang & Agrawala (2023); Yang et al. (2023) aims to transform structural inputs, *e.g.*, mask, edge or pose, to photorealistic images according to exemplars' content. Park *et al*. Park et al. (2019) trained SPADE, in which exemplars' global styles were captured by an encoder and were passed to the spatially-adaptive normalization to synthesize images. A better style control was proposed in CoCosNet Zhang et al. (2020), constructing dense semantic correspondence between a structural input and its exemplar. Zhou *et al*. Zhou et al. (2021a) improve CoCosNet by leveraging a hierarchical PatchMatch method to fast compute full-resolution correspondence. Recent progress has also been made to introduce unbalanced optimal transport Zhan et al. (2021), automatic assessment Guo et al. (2022), contrastive learning Zhan et al. (2022), and dynamic sparse mechanism Liu et al. (2022) for effective semantic correspondence learning. Meanwhile, techniques for diffusion models Seo et al. (2022); Bhunia et al. (2023); Zhang & Agrawala (2023); Huang et al. (2023); Yang et al. (2023); Goel et al. (2023); Wang et al. (2023) also step into this field, in which Zhang & Agrawala (2023); Mou et al. (2023) setup adaptive encoders for diffuser, Huang et al. (2023) disentangle and reassemble attributes of images, Yang et al. (2023) use CLIP image encoding for inpainting, Goel et al. (2023) utilize semantic masks, and Wang et al. (2023) uses all types of conditional inputs including images.

## 3 METHOD

### 3.1 PRELIMIARIES

**Diffusion process** $q(x_T|x_0)$ and $p_\theta(x_T|x_0)$ are $T$-step Markov Chains Ho et al. (2020) that gradually degrade $x_0$ to $x_T$ with random noises and recover $x_T$ from these noises:

$$q(x_T|x_0) = \prod_{t=1}^{T} q(x_t|x_{t-1}) = \prod_{t=1}^{T} \mathcal{N}(\sqrt{1-\beta_t}x_{t-1}; \beta_t \mathbf{I})$$
$$p_\theta(x_{t-1}|x_t) = \mathcal{N}(\mu_\theta(x_t,t), \Sigma_\theta(x_t,t))$$

in which $\beta_t$ is the standard deviation of the mixed-in noise at step $t$, $\mu_\theta(x_t,t)$ and $\Sigma_\theta(x_t,t)$ are network predicted mean and standard deviation under parameter $\theta$ of the denoised signal at step $t$. The loss function of training is the variational bound for negative log-likelihood Ho et al. (2020) shown as the following:

$$L = \mathbb{E}[-\log p_\theta(x_0)] \leq \mathbb{E}\left[-\log \frac{p_\theta(x_{0:T})}{q(x_{1:T}|x_0)}\right]$$

**CLIP** Radford et al. (2021) is a double-encoder network that bridges text-image pairs by minimizing the contrastive loss (*i.e.* cosine-similarity) between their embeddings. CLIP has served as an important prior module for nowadays T2I models such as DALLE-2 Ramesh et al. (2022) and Stable Diffusion Rombach et al. (2022). Also, it had been proved by prior works Ramesh et al. (2021; 2022) that its well-aligned cross-modal latent space is one of the core reasons for the T2I models' success.

**Image-Variation** defines a task that generates images with similar high-level semantics according to another image Ramesh et al. (2022); Xu et al. (2022). Prompt-Free Diffusion is closer to Image-Variation than exemplar-based generation approaches, in which the former finetunes existing T2I models using CLIP image encoder with frozen weights, while the latter proposed domain-specific networks for tasks such as virtual try-on, makeup, *etc*.

### 3.2 PROMPT-FREE DIFFUSION

As mentioned in earlier sections, we aim to propose an effective solution to handle nowadays high-demanding personalized T2I while aggressively maintaining *all merits* from prior approaches, *i.e.* high-quality, training-free, and reusable to most open-sourced models. Table 1 explains the pros and cons of varieties of approaches, in which we gauge from three angles: personalization quality, easy installation & domain adaptation, and input complexity & flexibility. The design of Prompt-Free Diffusion inherits T2I and Image-Variation models, consisting of a diffuser and a context encoder as two core modules, as well as an optional VAE that reduces dimensionality in diffusion. Particularly in this work, we have kept a precise latent diffusion structure like Stable Diffusion Rombach et al. (2022), shown in Figure 2.

| Methods *vs*. Properties | Personalization Quality | Easy Installation & Domain Adaptation | Input Complexity & Flexibility |
| --- | --- | --- | --- |
| Model Finetuning (DreamBooth Ruiz et al. (2023) *etc*.) | Full personalization | No easy installation & adaptation; Data and GPUs are required; Individual weights are required for individual domains | OK when special tokens are learned; Subject to input prompt quality |
| Prompt Searching & Engineering | Personalization is only available within a limited range, and is likely infeasible in complex cases | No installation required; Reusable to all T2I models & domains | Prompt searching & engineering is for sure required |
| Adaptive Layers with Structural Inputs (ControlNet Zhang & Agrawala (2023) *etc*.) | Users can customize the output structure but still has insufficient control over other aspects such textures, styles, backgrounds, *etc*. | Easy installation; Reusable to most T2I models & domains | Structural inputs such as depth and edges are required; Subject to input prompt quality |
| Image-Variation (Versatile Diffusion Xu et al. (2022) *etc*.) | Users can control only high-level semantics but has insufficient control over structures and others | Separate models are required; Not reusable to T2I models & domains. | Image inputs only; No prompts are needed |
| **Prompt-Free Diffusion (ours)** | Nearly full personalization; Users may control output structures, textures, and backgrounds using conditional image inputs | Easy installation by replacing CLIP with SeeCoder; Reusable to most T2I models & domains | Structural inputs such as depth and edges are optional; Image inputs only; No prompts are needed |

Table 1: This table compares the pros (green) and cons (red) from different methodologies with our Prompt-Free Diffusion along three aspects: Personalization quality; Easy installation & domain adaptation; Input complexity & flexibility. Yellow represents neutral.

Recall that text prompts are first tokenized and then encoded into $N$-by-$C$ context embeddings using CLIP in common T2I. $N$ and $C$ represent the count and dimension of the embeddings. Later,

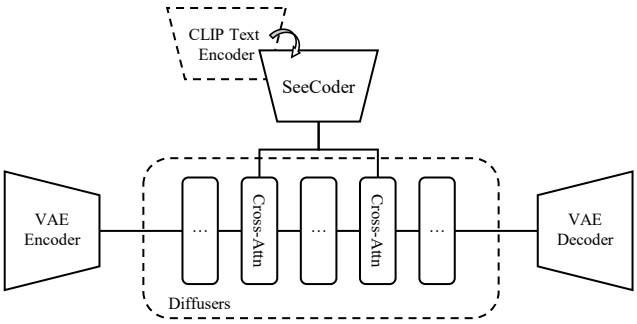

Figure 2: Graphic illustration of our Prompt-Free Diffusion with latent diffusion pipeline in which CLIP text encoder is replaced by the newly proposed SeeCoder.

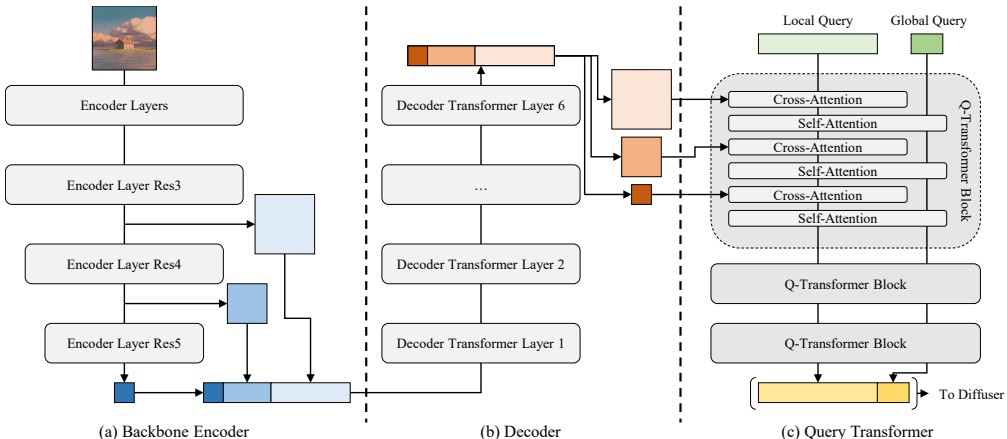

Figure 3: The overall structure of our proposed *Semantic Context Encoder (SeeCoder)*, which includes a *Backbone Encoder*, a *Decoder*, and a *Query Transformer*. For simplicity, we hind several detail designs: lateral connections between encoder and decoder; learnable query, level, and 2D spatial embeddings.

these embeddings are fed into the diffuser's cross-attention layers as inputs. In our Prompt-Free Diffusion, we replace the CLIP text encoder with the newly proposed SeeCoder (see Figure 2). Instead of text prompts, SeeCoder is designed to take image inputs only. It captures visual cues and transforms them into compatible $N$-by-$C$ embeddings representing textures, objects, backgrounds, *etc*. We then proceed with the same cross-attention layers in diffusers. Our Prompt-Free Diffusion also doesn't need any image disentanglement as priors (such as in Composer Huang et al. (2023)) because SeeCoder can determine a proper way to encode low- and high-level visual cues in an unsupervised manner. For more details, please see Section 3.3.

## 3.3 SEMANTIC CONTEXT ENCODER

As the core module in Prompt-Free Diffusion, our Semantic Context Encoder (SeeCoder) aims to take only image inputs and encode all visual cues into an embedding. One may notice that CLIP may also encode images. But in practice, CLIP's ViT Dosovitskiy et al. (2020) shows limited capacity because a) unable to take inputs higher than resolution $384^2$; b) does not capture detail textures, objects, *etc*.; c) trained with contrastive loss making it an indirect way of processing visual cues. Therefore we propose SeeCoder, a solution better fits vision tasks than CLIP.

SeeCoder can be breakdown into three components: *Backbone Encoder*, *Decoder*, and *Query Transformer* (see Figure 3). **Backbone Encoder** uses SWIN-L Liu et al. (2021) because it transforms arbitrary-resolution images into feature pyramids that better capture visual cues in different scales. For the **Decoder**, we proposed a transformer-based network with several convolutions. Specifically speaking, the Decoder takes features from different levels; uses convolutions to equalize channels; concatenates all flattened features; then passes it through 6 multi-head self-attention modules Wang et al. (2018) with linear projections and LayerNorms Ba et al. (2016). The final outputs are split and shaped back into 2D, then sum with lateral-linked input features.

The last part of SeeCoder is **Query Transformer** which finalizes multi-level visual features into a single 1D visual embedding. The network started with 4 freely-learning global queries and 144 local queries. It holds a mixture of cross-attention and self-attention layers, one after another. The cross-attentions take local queries as $Q$ and visual features as $K$ and $V$. The self-attentions use the concatenation of global and local queries as $QKV$. Such design prompts a hierarchical knowledge passing in which the cross-attentions transit visual cues to local queries, and self-attentions distill local queries into global queries. Besides, the network also contains free-learned query embeddings, level embeddings, and optional 2D spatial embeddings. The optional spatial embeddings are sine-cosine encodings followed by several MLP layers. We name the network SeeCoder-PA when there exist 2D spatial embeddings, where PA is short for Position-Aware. Finally, global and local queries are concatenated and passed to the diffuser for content generation. Notice that our network shares similarities with segmentation approaches, such as Jain et al. (2023); Li et al. (2022); Cheng et al. (2022); Jain et al. (2021). Nevertheless, their end purposes vary: to capture visual embeddings for discriminative *vs*. generative tasks.

## 4 EXPERIMENTS

### 4.1 DATA AND TRAINING

Aligned with many prior works Ramesh et al. (2022); Rombach et al. (2022); Xu et al. (2022), we adopted Laion2B-en Schuhmann et al. (2021) and COYO-700M Byeon et al. (2022) as our training data. Laion2B and COYO are large-scale image-text pairs that contain 2 billion and 700 million web-collected samples. Both datasets were frequently used in T2I research. Since Prompt-Free Diffusion requires no prompts, we actually only used these datasets' image collections for model training and evaluation.

The pretrained models' selection impacts Prompt-Free Diffusion's final model due to its unique training procedure with frozen weights. As mentioned in Section 3.3, we chose SWIN-L Liu et al. (2021) as SeeCoder's Backbone Encoder, and SD2.0's Rombach et al. (2022) as VAE. We used our in-house T2I diffuser, which outperformed SD1.5 as the pretrained diffuser for Prompt-Free Diffusion. Despite involving our in-house model, we demonstrated that a well-trained SeeCoder is reusable to other open-source T2I models in Section 4.3.

Other training settings are listed as the following: We used DDPM with $T = 1000$ diffusion timesteps with linearly increased $\beta$ from $8.5 \times 10^{-5}$ to $1.2 \times 10^{-2}$. For each iteration, we sampled DDPM timestep $t \in T$ uniformly. We trained the model with 100k iterations, 50k with a learning rate $10^{-4}$, and the other 50k with $10^{-5}$. Our training batch size was set to 512, 8 samples per GPU, a total of 16 A100 GPUs across two nodes, and a gradient accumulation of 4.

Besides, we also train a separate position-aware model with 2D spatial embeddings, namely SeeCoder-PA. Prompt-Free Diffusion with SeeCoder-PA performs better than SeeCoder when no structural conditionings are used (see Sec 4.2), an explainable phenomenon as the spatial embeddings partly cover the missing structural inputs. SeeCoder-PA is trained from a 50k SeeCoder checkpoint and finetuned additional 20k steps with a learning rate $5 \times 10^{-5}$.

### 4.2 PERFORMANCE

We demonstrate the performance of Prompt-Free Diffusion in Figure 4, in which our method generates high-quality images replicating details from reference inputs. In this experiment, Prompt-Free Diffusion extensively uses ControlNets Zhang & Agrawala (2023) to handle a variety of structural conditionings, including but not limited to canny-edge, depth, mlsd, and scribble. Also, Prompt-Free Diffusion is insensitive to input resolution and aspect ratios. Examples are shown in Figure 4 with three scales: $512^2$, $512 \times 768$, and $768 \times 512$. The reference dimension *does not* need to match the output dimension, Figure 4 shows cases with matched dimensions merely for better exhibition. Recall that in Section 4.1, we mentioned that an in-house T2I diffuser that outperforms SD1.5 are used in training. Such a diffuser is also used to generate the results in this experiment and all following experiments unless separately mentioned.

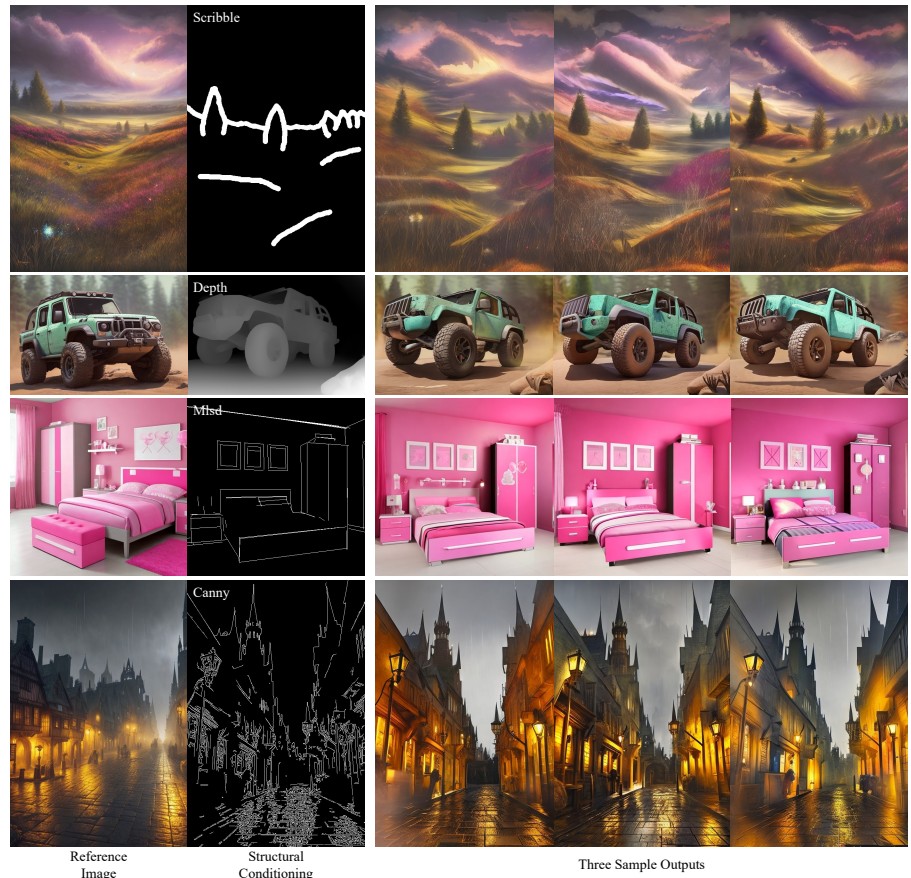

Reference Image     Structural Conditioning     Three Sample Outputs

Figure 4: This figure shows results from Prompt-Free Diffusion, in which we sample three outputs for each case using one reference image and one structural conditioning (*i.e.* canny edge, depth, mlsd, and scribble) as input. No prompts are required.

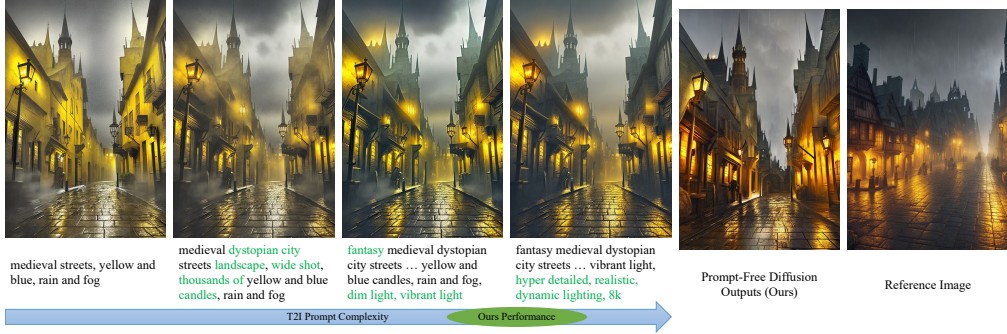

Figure 5: This figure gathers the performance of ControlNet+T2I using prompts with progressive complexity. It starts with the prompt *"medieval streets, yellow and blue, rain and fog"* which gives basic semantics and styles. Later we improve the performance with semantic decorative prompts such as *"dystopian"* and *"wide shot"*, style prompts such as *"fantasy"*, and common prompt engineering such as *"realistic"* and *"8k"* (displayed in green). Our Prompt-Free Diffusion matches the quality of the top two most sophisticated prompts.

**Compare with T2I:** The following experiment shows the performance comparison between our Prompt-Free Diffusion and the traditional prompt-based ControlNet+T2I Zhang & Agrawala (2023); Rombach et al. (2022) (see Figure 5). Specifically, we use prompt inputs with progressive complexities and check which level of prompt complexity is equivalent to our Prompt-Free Diffusion in performance. As shown in Figure 5, our approach roughly reaches the top two levels: between "requiring semantic and style decorative prompt" and "requiring extra prompt engineering". We also noticed a tricky color shift using ControlNet+T2I, which we gave up fixing after numerous tries.

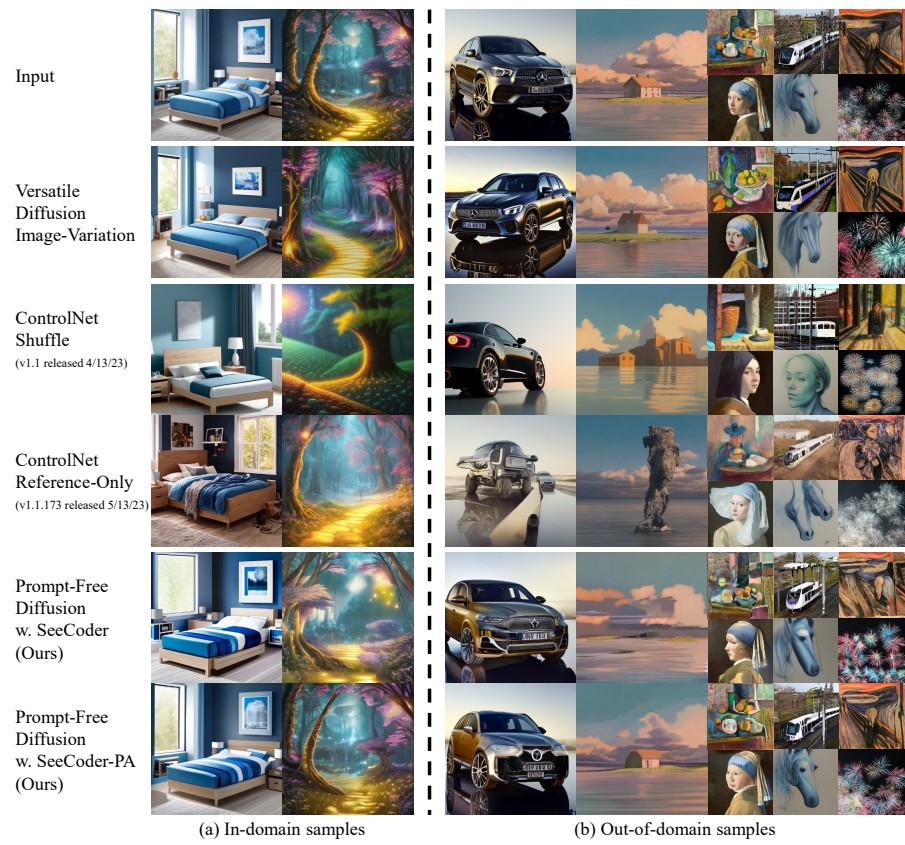

|  | (a) In-domain samples | (b) Out-of-domain samples |
|---|---|---|

Figure 6: Image-Variation comparison between VD Xu et al. (2022), ControlNet Zhang & Agrawala (2023), and Prompt-Free Diffusion. Testing samples s are categorized into a) in-domain and b) out-of-domain, meaning whether the input can be generated using the pretrained T2I diffuser. In conclusion, Prompt-Free Diffusion w. SeeCoder-PA beats ControlNet Shuffle and Reference-Only for both in-domain and out-of-domain cases.

**Image-Variation:** Next, we evaluate Image-Variation (IV): generating images from other reference images. Notice that IV is a natural setup for Prompt-Free Diffusion when no ControlNet is involved. Prior baseline such as VD Xu et al. (2022) finetunes a diffusion model for IV. Concurrently, the ControlNet team also proposed two new models, "shuffle" and "reference-only" (v1.1.173) Zhang & Agrawala (2023), so we compared both. Testing samples are categorized into in-domain and out-of-domain, meaning whether the reference images were generated by the pre-trained T2I diffuser, or from a different source. We draw the following conclusions in this test: a) VD has the best overall performance, meaning finetuning still yields good performance; b) ControlNet Shuffle and Reference-Only have significant performance drops in out-of-domain tests. Reference-Only outperforms Shuffle on in-domain samples and vice versa on out-of-domain samples. These results reflect ControlNet's weakness in generality and show its limitation of not having quality and generality in both hands. c) Prompt-Free Diffusion better replicates the reference images (*i.e.* semantic, texture, background, *etc.*), which aligns with its training objective. SeeCoder-PA (*i.e.* SeeCoder with spatial encoding) outperforms SeeCoder in terms of quality (see Section 4.1). Both models also do well on in-domain samples and have quality gaps on out-of-domain samples. Overall, our approach beats ControlNet for Image-Variation.

## 4.3 REUSABILITY AND DOWNSTREAM APPLICATIONS

**Reusability** is a critical property of our SeeCoder. We test the property by *directly plugging in a pre-trained SeeCoder* with six other open-source models, including the base model *SD1.5* Rombach et al. (2022); art-focused models *OpenJourney-V4* and *Deliberate-V2*; anime models *AOM-V2* and *Anything-V4*; and the photorealistic model *RealisticVision-V2*. We emphasize that the pre-trained SeeCoder is frozen in those cases - no re-training is needed. The results are shown in Figure 7, demonstrating that SeeCoder is highly reusable to other T2I models.

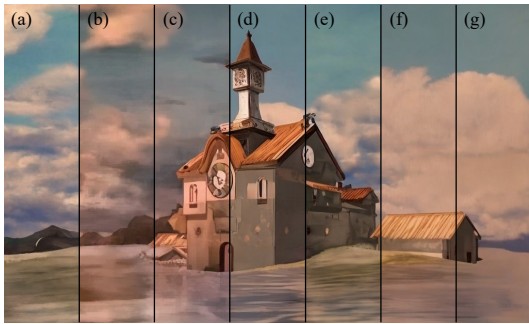

Figure 7: SeeCoder's adaptability examination on 7 open-sourced & in-house models: (a) SD1.5 Rombach et al. (2022); (b) AOM-V2; (c) Anything-V4; (d) In-house model (*i.e.* the diffuser SEE was trained with); (e) OpenJourney-V4; (f) Deliberate-V2; (g) RealisticVision-V2
.

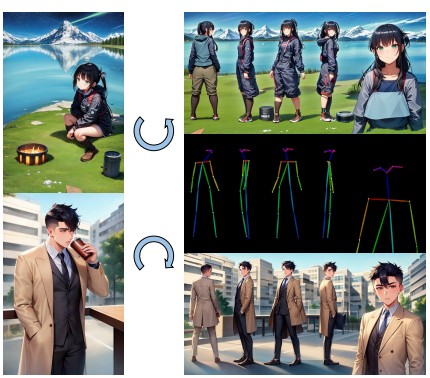 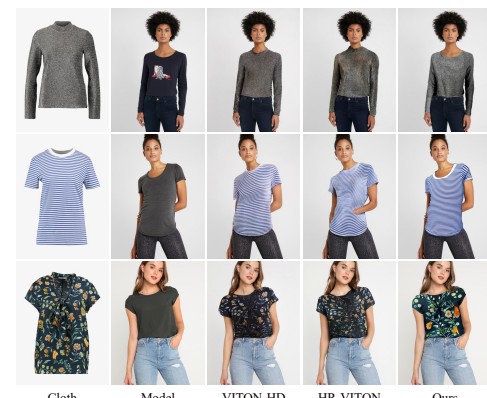

Figure 8: Left: Anime figures generated by Prompt-Free Diffusion with reference images and conditioning poses. Right: Virtual try-on using Prompt-Free Diffusion compared with other state-of-the-art approaches.

**Image-based Anime Figure Generation** is one of the practical uses of Prompt-Free Diffusion in anime or game design. In Figure 8, we generate anime figures based on SeeCoder-captured visual cues and conditional poses. To better accommodate anime data that is "out-of-domain", SeeCoder here is finetuned on AOM-V2 and its synthesized images for 30k iterations. From the results, Prompt-Free Diffusion demonstrates promising results for this task.

**Virtual Try-on** is an exemplar-based generation task explored by works such as Choi et al. (2021) and Lee et al. (2022). Prompt-Free Diffusion can also handle this task with minor modifications: We obtain cloth masks from SAM Kirillov et al. (2023) and feed them to ControlNet. Simultaneously, we use SeeCoder's visual embedding to guide our generation. Results are shown in Figure 8. Other tasks such as model cognitive study Zhang et al. (2023) and video generation Khachatryan et al. (2023); Singer et al. (2022); Blattmann et al. (2023) may also supported by SeeCoder.

## 5 CONCLUSION AND ETHICAL DISCUSSION

In this article, we propose Prompt-Free Diffusion, a novel pipeline that generates personalized outputs based on exemplar images, not text prompts. Through experiments, we show that our core module, SeeCoder, can generate high-quality results and can easily plug-and-use in various well-established T2I pipelines through CLIP replacement. Last but not least, SeeCoder shows great potential in handling practical tasks such as anime figure generation and virtual try-on with surprising quality, making it a further solution for downstream users.

While our proposed Prompt-Free Diffusion can assist artists and designers in creative content generation, it is important to acknowledge that the misuse or abuse of our system may cause negative social impacts and ethical concerns similar to other controllable image synthesis approaches. In addition, open-source pretrained text-to-image models used in conjunction with Prompt-Free Diffusion may contain harmful bias, stereotypes, *etc*. As a crucial step to address these concerns, we encourage users to deploy and utilize our approach in a responsible way with ethical regulations and enhanced transparency.

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
