# OpenReview forum: "Prompt-Free Diffusion: Taking “Text” out of Text-to-Image Diffusion Models"
_ICLR.cc/2024/Conference — ICLR 2024 Conference Withdrawn Submission_

### Official Review · Reviewer_tGkr · 2023-10-29

**Soundness:** 2 fair
**Presentation:** 3 good
**Contribution:** 2 fair
**Rating:** 5
**Confidence:** 5

**Summary:**

This paper propose  Semantic Context Encoder (SeeCoder) to replace the function of CLIP text encoder to achieve image-to-image variation. More specifically, they trained a transformer-based model to replace the CLIP in stable diffusion. They show good performance on image variation task.

**Strengths:**

1. The proposed method is intuitively sound. Training an image encoder to align with the stable diffusion to achieve image variant.
2. Also, following intuition, it works on diffusion model based on stable diffusion such as some ControlNet checkpoint, which expands the results.
3. They demonstrate good perfomance on image variant tasks.

**Weaknesses:**

1. The image encoder requires further training, while methods like MasaCtrl could achieve image variants without help from an extra image encoder. MasaCtrl is definitely a missing baseline.
2. Many other methods like AnyDoor: Zero-shot Object-level Image Customization achieved great performance in learning image representation. For example, AnyDoor also does virtual try-on tasks.
3. The author should do an ablation on Seecoder since this is the main structure novelty. Also, if that structure is important, they should compare it with transformer-based block. Without further investigation exhibited in the paper, I am not sure if such a specifically designed Seecoder is necessary. Is it possible that a vision transformer block with a convolution to output feature NxC also works?

**Questions:**

1. Why using such design in Seecoder? Have you tried other structure?
2. How does it compare with other methods?
3. How does it perform for general images when no cherry-pick is used?

---

### Official Review · Reviewer_9Xqi · 2023-10-31

**Soundness:** 2 fair
**Presentation:** 1 poor
**Contribution:** 2 fair
**Rating:** 3
**Confidence:** 3

**Summary:**

The paper introduces a diffusion-based, "prompt-free" image generation model. The main contribution is its design of the "Semantic Context Encoder", abbreviated as "SeeCoder", which is a "reusable" image encoder that can be plugged into trained diffusion-based image generation models. The proposed model takes no text input, but image input ("structural conditioning") that is scribble, depth, canny edge etc.

**Strengths:**

* The authors test their model on a wide range of image generation tasks, including conditional image generation, anime figure generation, and virtual try-on. The results appear competitive against the state of the arts.

**Weaknesses:**

* The paper's motivation is not convincing. The paper tries to replace text prompts with image-based "prompts", as text prompt is ambiguous and less controllable. However, in many of the presented examples, the image prompts cannot be generated without using a specialized tool - Producing depth map, canny edge, human pose skeleton all require running certain program on *existing image*. In addition, it is hard to produce a depth map or canny edge without having the target image ready, and they cannot be easily altered. Therefore, the proposed image prompt appears neither as accessible nor as controllable as text prompt.

* The paper's description on the method and the experiments are oversimplified and vague. In particular, I find little explanation on how the method achieves training-free reusability, which is a key point of this paper. I also find little information on how the model comes to understand the various types of "structural conditioning", i.e. image prompt. Many such questions remain after reading the paper and are listed below. They severely hurt the readability of this paper.

**Questions:**

- In Section 3.3, the "Decoder" is described as a "transformer-based network with several convolutions (*where?*), ... uses convolution to equalize channels (*whose?*), ... 6 multi-head self-attention modules". The model looks like a Transformer encoder rather than decoder - otherwise, what are the query and memory?

- Also in Section 3.3, the "Query Transformer ... started with 4 freely-learning global queries and 144 local queries", What exactly are these queries? Where "local" comes from? "The network also contains free-learned query embeddings, level embeddings, and optional 2D spatial embeddings". I could not find the definition of these in the text or in any illustration.

- As aforementioned, how the method achieves training-free reusability?

- As aforementioned, how the model is trained to understand, and hence follows the instruction of, image prompts that comes in various forms? Are the image prompts illustrated in the paper, including depth map, canny edge, scribble, also in the training data?

---

### Official Review · Reviewer_ESh9 · 2023-11-01

**Soundness:** 2 fair
**Presentation:** 2 fair
**Contribution:** 2 fair
**Rating:** 1
**Confidence:** 5

**Summary:**

This paper proposes an image generation method, called Prompt-Free Diffusion. It relies on only visual inputs to generate new images, and removes the text prompts from the image generation process. Also, to extract the rich visual information from input images, the authors devise a visual encoding module Semantic Context Encoder (SeeCoder).

**Strengths:**

I cannot find any strength from this work.

**Weaknesses:**

1. The motivation of this work is questionable. Why do you want to taking “text” out of text-to-image diffusion models? Why not keep compatibility with "text"? Take ControlNet as an example, you can add more conditions (such as Canny edges and depth maps) into the framework, while allowing text prompts. Eliminating text prompts will simply limit the potential abilities of the model. When you do not need a text prompt, it is fine to just set it as an empty string.
2. Only several groups of qualitative examples are presented, while NO quantitative results are given.
3. The effectiveness of the proposed Semantic Context Encoder (SeeCoder) module is not fully validated. The authors should conduct more experiments and ablation study to prove its value.
4. The authors claimed that they used a in-house T2I diffuser, which outperformed SD1.5, but did not provide any convincing evidence. This is not acceptable.

**Questions:**

The authors should resolve the concerns in the Weaknesses section.

---

### Official Review · Reviewer_KASv · 2023-11-01

**Soundness:** 2 fair
**Presentation:** 1 poor
**Contribution:** 1 poor
**Rating:** 3
**Confidence:** 4

**Summary:**

This paper discusses the rapid growth of text-to-image (T2I) research, driven by large pre-trained models and personalization techniques. However, a challenge remains in crafting effective text prompts for customized results. To address this issue, the authors propose a novel framework called "Prompt-Free Diffusion" that eliminates the need for text prompts in T2I models. This framework relies solely on visual inputs, using a Semantic Context Encoder (SeeCoder) instead of text encoders like CLIP or LLM. Through experiments, Prompt-Free Diffusion is shown to outperform prior methods, match state-of-the-art T2I models, and extend to applications like anime figure generation and virtual try-on. The authors plan to open-source their code and models.

**Strengths:**

The authors have dealt with an interesting problem. The problem statement is well-defined but the novelty/contributions are poor.

**Weaknesses:**

- The novelty of this work is very limited. This seems to be a naive architectural of ImageVariation work.
- The methodology is not nicely written.
- Fig. 3 diagram should have been more polished and crisper.

**Questions:**

What is the difference between Image-Variation and the proposed method -- besides replacing CLIP Image encoder with SeeCoder (consisting of Backbone Encoder, Decoder, and Query Transformer)?